# Diversity and Antimicrobial Activity of Culturable Fungal Endophytes in *Solanum mauritianum*

**DOI:** 10.3390/ijerph17020439

**Published:** 2020-01-09

**Authors:** Sharon Pelo, Vuyo Mavumengwana, Ezekiel Green

**Affiliations:** 1Department of Biotechnology and Food-Technology, University of Johannesburg, 55 Beit Street, Doornfontein, Johannesburg 2028, South Africa; shanon.pelo@gmail.com; 2DST-NRF Centre of Excellence for Biomedical Tuberculosis Research, South African Medical Research Council Centre for Tuberculosis Research, Division of Molecular Biology and Human Genetics, Faculty of Medicine and Health Sciences, Stellenbosch University, Tygerberg Campus, Cape Town 7505, South Africa; vuyom@sun.ac.za

**Keywords:** antimycobacterial, secondary metabolites, fungal endophytes, minimum inhibitory concentration, *Mycobacterium tuberculosis*, *Solanum mauritianum* and *Penicillium chrysogenum*

## Abstract

Plant endophytes are microbial sources of bioactive secondary metabolites, which mimic the natural compounds chemistry of their respective host plants in a similar manner. This study explored the isolation and identification of fungal endophytes, and investigated the antibacterial and antimycobacterial activity of their crude extracts. Fungal endophytes were isolated from *Solanum mauritianum*, identified using morphological traits and internal transcribed spacer ribosomal-deoxyribonucleic acid (ITS-rDNA) sequence analysis. Eight fungal endophytes were identified as *Aureobasidium pullulans, Paracamarosporium leucadendri*, *Cladosporium* sp., *Collectotrichum boninense, Fusarium* sp., *Hyalodendriella* sp., and *Talaromyces* sp., while *Penicillium chrysogenum* was isolated from the leaves and unripe fruits. Good activity was observed for the crude extracts of *Paracamarosporium leucadendri* inhibiting *Mycobacterium bovis*, *Klebsiella pneumoniae,* and *Pseudomonas aeruginosa* at 6 µg/mL. Crude extracts of *Fusarium* sp., showed activity at 9 μg/mL against *M. bovis, M. smegmatis* and *K. pneumonia*. In general, the crude extracts showed great activity against Gram-negative and Gram-positive bacteria and novel results for two mycobacteria species *M. bovis* and *M. smegmatis*. The results provide evidence of diverse fungal endophytes isolated from *Solanum mauritianum,* and evidence that fungal endophytes are a good source of bioactive compounds with pharmaceutical potential, particularly against *Mycobacterium tuberculosis.*

## 1. Introduction

Plants act as hosts to several endophytic microorganisms, known to produce bioactive compounds of novel potential use in agriculture, modern medicine and the pharmaceutical industry. Endophytes have a symbiotic association with the host plant [1] and are responsible for assisting the plants to adapt to abiotic and biotic stress [2]. Endophytes include bacteria and fungi, however fungal endophytes are the most researched because they are a potential source of bioactive natural products and have been shown to produce a variety of secondary metabolites with interestingly complex molecular scaffolds [3,4,5].

Several metabolites of interest have been isolated from fungal endophytes, including novel compounds with antibiotics, antialgal, anticancer, antifungal, antimalarial and immunosuppressive activities [6], for example, the anticancer drug paclitaxel (taxol) was isolated from *Pestalotiopsis microspora*, an endophytic fungus that colonizes the *Taxus wallichiana* tree [7], and camptothecin, a topoisomerase inhibitor, has been isolated from *Fusarium solani* endophytic fungi in *Camptotheca acuminata* [8]. Vinblastine and Vincristine have also been isolated from the endophytic fungus *Fusarium oxysporum*, from *Catharanthus roseus* [9]. Recent reports have revealed that endophytes produce the same secondary metabolites as those of the host plant, making them a promising source of novel compounds [10].

Fungal endophytes in *Panax ginseng* have been investigated for their ability to produce ginsenosides [11], and the steroidal alkaloid Solamargine from fungal endophytes of *Solanum nigrum* L. [12]. Solamargine has been reported to be cytotoxic and has been experimentally attempted on various cancer cell lines (related to skin, breast and liver) [13]. In this study, efforts have been made to isolate endophytic fungi inhibiting a range of pathogenic bacteria including *Mycobacterium bovis*.

## 2. Materials and Methods 

### 2.1. Sampling Method

Fresh leaves, stems and fruits from healthy specimens of *Solanum mauritianum* were collected from the campus garden at the University of Johannesburg (26.11°32.6′ S, 28.03°28.9′ E), in Johannesburg, South Africa. The plant species was chosen based on it being invasive, as it grows fast with massive encroachment into native vegetation (hence, its extermination in South Africa). The plant was identified by a botanist at the University of Johannesburg. Voucher specimen (BTNPSP02) of the plant was deposited in the herbarium of the University of Johannesburg. Isolation of fungal endophytes was carried out within 24 h of collection, in the Molecular Pathogenesis and Molecular Epidemiology Research Group (MPMERG) laboratory of the Biotechnology and Food Technology Department. 

### 2.2. Plant Sterilization and Endophytes Isolation

Endophytes were isolated from different parts of the plant, as explained by Huang et al. [14]. Surface sterilization was performed following the procedure of Larran et al. [15]. Briefly, plant parts were washed thoroughly in running tap water, immersed in 250 mL of 70% ethanol for 1 min, then in 250 mL of 5% sodium hypochlorite solution for 5 min, and rinsed in 3 L of sterile distilled water three times separately. The sterile surface sample was cut into 5 mm pieces with a sterile blade, inoculated onto 90 mm plates different agar including Potato Dextrose Agar (PDA), Sabouraud Dextrose Agar (SDA) and Nutrient agar and incubated at 25 °C for 7–14 days in the dark. After 7 days, emerging hyphae were transferred to fresh plates for subculturing into pure isolates and subsequent identification. Morphological identification of the isolates was performed according to Katoch et al. [16].

### 2.3. Molecular Identification of Fungal Endophytes

The pure colonies of fungal isolates were subjected to a molecular identification method described by White et al. [17]. Nine morphotypes were selected for further molecular identification and phylogenetic analysis, namely SPS (Sharon Pelo Sample) 28, SPS 33, SPS 35, SPS 38. SPS 39, SPS 40, SPS 41, SPS 42 and SPS 43. Pure cultures were sent to Inqaba Biotec Africa’s Genomics Company (Pretoria, South Africa) for molecular identification. The method of Stephen et al. [18] was used for DNA isolation. The ITS target region was amplified according to Stephen et al. [18] using two primers, ITS1 (5′ to 3′) (TCCGTAGGTGAACCTGCGG) and ITS4 (5′ to 3′) (TCCTCCGCTTATTGATATGC). The PCR products were gel extracted with Zymo Research, Zymoclean™ Gel DNA Recovery Kit (Zymo Research, Irvine, CA, USA), and sequenced in the forward and reverse directions on the ABI PRISM™ 3500xl Genetic Analyzer (ThermoScientific, City Waltham, MA, USA). Purified sequencing products were cleaned using Zymo Research, ZR-96 DNA Sequencing Cleanup Kit™ (Zymo Research, Irvine, CA, USA) and were then analyzed using CLC Main Workbench 7 followed by a BLAST search (NCBI). Representative sequences for each isolated fungal endophyte were deposited in the GenBank. The taxa that had 99% ITS sequence similarities were identified as reference taxa species, and some of the described species were identified at the genus level. 

### 2.4. Phylogenetic Analysis

The sequence data obtained from Inqaba biotect were aligned using Edit Sequence Alignment Editor v.7.2.3 software [19]. The search for homologous sequences was done using Basic Local Alignment Search Tools (BLAST) at the National Center for Biotechnology Information online [20] All fungal endophyte sequences with 99–100% similarities had the best hit in the NCBI database [21]. Alignments of nucleotide sequences (isolate and species obtained through BLAST) were performed using MUSCLE with default options. Phylogenetic trees were constructed using a Maximum likelihood method (ML) method based on the Kamura-Nei model and bootstrap test [22] of 1000 replicates. 

### 2.5. Preparation of the Crude Extracts 

The fungal endophytes that were grown on PDA and cut into 10 mm plugs were placed in a 200 mL short bottle containing 100 mL of Potato dextrose broth (PDB) and cultured for 14 days at 25 °C ± 2 °C under static conditions [23]. After fermentation, the mycelium that formed on top was macerated with a mortar and pestle, then filtered on Whatman 1 filter paper. Afterwards, 100 mL of ethyl acetate was added to the bottle, then left at room temperature for 24 h. The organic solvent phase was transferred into a round bottom flask for concentration under vacuum using a rotary evaporator (Labtech EV311H) at 40 °C. The extraction process was repeated 3 times. The extracted secondary metabolites were poured into a McCartney bottle and left under a laminar flow to dry. The dried mycelia (and spores) were further extracted in 100 mL of methanol and left at room temperature for 24 h. These were filtered and the methanol extracts were evaporated at 40 °C at 200 rpm. 

### 2.6. Mycobacterium Inoculum Preparation

*Mycobacterium bovis* (ATCC 27290) and *Mycobacterium smegmatis* (ATCC 607) were prepared by inoculating the bacterium into a freshly prepared 10 mL Middlebrook 7H9 broth base, supplemented with Middlebrooks OADC (Oliec albumin Dextrose Catalase) growth supplement (Sigma-Aldrich, St. Louis, MO, USA), and incubating at 37 °C for 7 days and 37 °C for 24 h, respectively. 

After the incubation period, the test tubes were vortexed and allowed to settle. The supernatant was transferred into Mueller-Hinton broth and adjusted using a spectrophotometer to a 0.5 McFarland standard and further diluted to1:10 in Mueller-Hinton broth for the test analysis. The other pathogenic strains were prepared according to the method of Marcellano at el. [23]. Briefly, the bacterial cultures were sub-cultured onto Mueller-Hinton broth and incubated overnight at 37 ℃. The overnight bacterial suspensions were adjusted to 0.5 McFarland standard for the test analysis.

### 2.7. Disc Diffusion Assay of Crude Extracts from Fungal Endophytes

The aliquot of the dried extract was dissolved in 10% dimethyl sulfoxide (DMSO). Concentrations of 10 mg/mL for each fungal endophyte crude extracts were prepared and inoculated into 6 mm discs and allowed to dry under sterile conditions for 1 min. Ten percent DMSO was used as a negative control in the bioassays, while rifampicin (40 µg/mL) was used as a positive control. Pathogenic bacteria such as *Bacillus subtilis* (ATCC 11774), *Escherichia coli* (ATCC 10536), *Klebsiella pneumonia* (ATCC 10031), *Staphylococcus aureus* (ATCC 6571), *Pseudomonas aeruginosa* (ATCC 10145), *Mycobacterium bovis* (ATCC 27290) and *Mycobacterium smegmatis* (ATCC 607) were grown on Mueller-Hinton agar (MHA) plates and used as test organisms. The plates were incubated at 37 ℃ for 24 h, while *M. bovis* plates were incubated for 7 days. Zones of inhibition were determined by measuring the diameters of the clear zones. The whole disc diffusion assay procedure was performed in triplicate.

### 2.8. Minimum Inhibitory Concentration Assay

The Resazurin Microtiter Assay (REMA) was performed according to Chien et al. [24]. The inoculum was prepared from fresh Mueller-Hinton agar (MHA), into Mueller-Hinton broth (MHB) and adjusted using a spectrophotometer to a 0.5 McFarland standard and further diluted to 1:10 in Mueller-Hinton broth for the test. Briefly, 100 μL of MHB was dispensed into each well of a flat bottom 96 well plate (Becton Dickinson, Franklin Lakes, NJ, USA). Serial dilution of the crude extract from fungal endophytes were prepared directly in the plate with concentrations ranging from 5 mg/mL to 0.0039 mg/mL. One hundred microliters of the test organism were added into each well. A negative control and positive control (ranging from 40 µg/mL to 0.019 µg/mL) were also included for each isolate. Sterile water was added to all perimeter wells to avoid evaporation during incubation. The plates were covered, sealed with a plastic bag and incubated at 37 °C under normal atmosphere. After 24 h of incubation, 30 µL of resazurin solution was added as described by Chien et al. [24]. Further incubation was required for *M. bovis* for 7 days, after which 30 μL of resazurin solution was added. A change of colour from blue to pink indicated the reduction of resazurin by proliferating bacteria. Minimum concentration was defined as the lowest concentration of the drug that prevented this change.

## 3. Results

### 3.1. Isolation and Identification

A total of eight endophytic fungi were isolated from *S. mauritianum*, with the control not showing any growth. Table 1 shows observed characteristics of the fungal endophytes. 

### 3.2. Molecular Confirmation

The results revealed a diversity of fungal endophytes from unripe fruits, ripe fruits, stem and leaves, as seen in Table 2. Two fungal isolates were isolated from unripe fruits while three were from leaves. 

### 3.3. Phylogenetic Analysis

The blast analysis of the 16S rRNA gene sequence resulted in 8 varying fungal genera (Figure 1). The results of phylogenetic analysis showed distinct clustering of the isolates at 94–100% similarities. Analysis of the ITS1 and ITS4 rDNA reveals that the most prevalent endophytic fungi was *Penicillium chrysogenum* (22.2%) isolated from the leaves and unripe fruit, *Fusarium* sp. (11.1%) and *Paracamarosporium leucadendri* (11.1%). The similarity levels of the sequences obtained ranged from 94–100%. 

### 3.4. Yield of Extracts

More extracts were obtained from *Cladosporium* sp. (1.02 g), *P. chrysogenum* (L) (1.24 g), and *P. leucadendri* (1.23 g) as compared to other endophytes in this study (Table 3).

### 3.5. Disc Diffusion Assay of Secondary Metabolites Extracts

All endophytes showed positive antimicrobial activity in the preliminary screening. All secondary metabolites from the nine fungal endophytes, with the exception of *Fusarium* sp., inhibited *Mycobacterium bovis* whilst only eight inhibited *Mycobacterium smegmatis.* The largest zone of inhibition was observed against *P. aeruginosa* (23 ± 2.7) inhibited by *Fusarium* sp. and *E. coli* and *K. pneumonia* (17.3 ± 2.1 mm, 19 ± 1.0 mm and 18.3 ± 0.6 mm) respectively, inhibited by *Talaromyces* sp. secondary metabolites extracted with methanol and ethyl acetate. Further activity was observed against *M. bovis,* by *Fusarium* sp. *at* (14 ± 1.7 mm) and *P. chrysogenum* (F) (14 ± 2.9 mm) extracts. *M. smegmatis* was inhibited by *P. leucadendri at* (13 ± 1.0 mm), while *S. aureus* was inhibited by *Talaromyces* sp. secondary metabolites. Each of the endophytic fungi produced bioactive compounds that exhibited antimicrobial activity against at least one test microorganism used, as shown in Table 4.

### 3.6. Minimum Inhibitory Concentration of the Crude Extracts

The MIC values ranged from 0.006 mg/mL to 8 mg/mL (Table 5). The crude extract of *P. leucadendri* (6 µg/mL) had the lowest MIC against *K. pneumoniae, P. aeruginosa* and *M. bovis*, while it showed the highest MIC value of 2.91 mg/mL against *M. smegmatis*. 

## 4. Discussion

In this study, fungal endophytes were isolated and identified from *Solanum mauritianum*, an invasive plant from South America and weed in South Africa that is facing extermination [25]. We did not find any previous studies on the isolation of fungal endophytes from *S. mauritianum*, and this is the first time we report a broad-spectrum antimicrobial activity against *Mycobacterium bovis* and *Mycobacterium smegmatis* from extracts of fungal endophytes isolated from *S. mauritianum*. The fungal diversity analysis reveals a mixed composition of the endophyte community from the leaves, ripe and unripe fruits and stem of *Solanum mauritianum*. Out of the nine fungal endophytes that were isolated from a healthy plant of *Solanum mauritianum*, *Penicillium chrysogenum* was isolated twice in different parts of the plant, with the unripe fruit and leaves’ endophytes showing different growth patterns. 

*Paracamarosporium leucadendri, Cladosporium* sp., *Collectotrichum* sp., *Fusarium* sp., *Hyalodendriella* sp., *Talaromyces* sp., and *Penicillium* sp. [11,26,27,28,29,30,31,32,33], have been identified as plant endophytes, proving that they are not plant pathogens. 

The results of the phylogenetic analysis show distinct clustering of the isolates at 94–100% similarities. Analysis of the ITS1 and ITS4 rDNA reveals that the most prevalent endophytic fungi was *Penicillium chrysogenum* (22.2%), isolated from the leaves and unripe fruits, *Fusarium* sp. (11.1%) and *Paracamarosporium leucadendri* (11.1%). The similarities of the sequences obtained ranged from 94–100%. 

Currently there is growing interest on the endophytic microorganisms, producing a wide range of compounds with diverse biological activities [34]. The results of the disc diffusion assay showed inhibition by all nine fungal endophyte crude extracts on all the test bacteria. There are many studies in literature on penicillin as the most important antibiotic and as the first discovered antibiotic in history [35,36,37,38] successfully used to treat bacterial infections and has been isolated from the fungus *P. chrysogenum* [35]. The endophytic fungi *P. chrysogenum* crude extract showed potential as an antibiotic with a broad-spectrum antimicrobial effect in this study. 

*Aureobasidium pullulans* was also isolated in our study; it has previously been reported to synthesize biomolecules of medicinal importance, for example, β-(1 → 3, 1 → 6)-glucans, which displays anti-tumour, immune-modulatory and food allergy inhibition [39], fungicide [40] and antimicrobial activity [41]. It also shows fungicide and fungicidal properties against *Candida albicans* [42]. The lowest MIC observed from *A. pullulans* was 0.17 mg/mL. *Hyalodendriella* sp., has been reported to contain bioactive properties that inhibit bacterial growth [28,43] and fungal growth [44]. This validates the results we obtained in this study, that fungal endophytes from plants have the potential to inhibit microorganisms other than bacteria, such as fungi.

Rodriguez et al. [45] reported *Fusarium* sp. to have resistance against virulent pathogens, as well as anticancer properties [45,46], anti-inflammatory, antitussive (to prevent or relieve a cough) and anti-allergic properties. In our study, *Fusarium* sp. secondary metabolite extracts showed greater activity on 4 microorganisms, including *M. bovis* and *M. smegmatis.* This is noteworthy as there are no reports on *Fusarium* sp. having antagonistic properties against mycobacteria.

The production of saponins by the fungal endophyte *Paracamarosporium leucadendri* has been previously reported [11]. Sharma et al. [47] reported the bioactivity of an ethyl acetate extract on two Gram negative and two Gram positive organisms at a concentration of 25 mg/mL. In our study we observed the activity of ethyl acetate extracts from *P. leucadendri* against *K. pneumonia, P. aeruginosa* and *M. bovis* at the lowest concentration of 6 µg/mL. These results, are indicative of yet unidentified active principals in the extract. 

Several bioactive secondary metabolites such as anthraquinone [48] and alkaloids [49] have been isolated and reported from *Talaromyces* sp., [50], with even newer compounds such as penicillic acid emerging due to co-culturing with *Fusarium solani* and other fungal endophytes [30]. In our study *Talaromyces* sp. showed inhibition at 30 µg/mL against *E. coli*, *K. pneumonia*, *P. aeruginosa*, *M. bovis* and *M. smegmatis*. The fungal endophyte *Penicillium chrysogenum* isolated from the leaves showed activity at the lowest MIC activity (10 µg/mL) against *Mycobacterium bovis* and *Mycobacterium smegmatis*, while *Fusarium* sp. showed activity at the lowest MIC against *Mycobacterium bovis* and *Mycobacterium smegmatis* (9 µg/mL) with *P. leucadendri* (6 µg/mL) showing the lowest MIC concentration against *Mycobacterium bovis*. This is encouraging because, according to Rios and Recio [51], natural product compounds that show antimicrobial activity at ≤10 µg/mL and 100 μg/mL for extracts should be considered noteworthy. As such, in this case, *A. pullulans*, *Cladosporium* sp., *Fusarium* sp., *Hyalodendriella* sp., *P. chrysogenum* (L), *P. chrysogenum* (F), *P. leucadendri,* and *Talaromyces* sp. have notable secondary metabolites.

It has been demonstrated that some compounds of plants origin are produced by fungal endophytes of that plant, and some of the biological activity of the plants can be found in their endophytes [50]. Fungi are known for their extraordinary contribution to drug production, as well as in managing diseases in animals and humans, as they produce a large variety of secondary metabolites of commercial and medicinal importance.

Ethyl acetate extracts of all fungal endophytes showed activities against *S. aureus*, *E. coli* and *M. bovis*, with zones of inhibition of 7 mm while that of *A. pullulans* on *B. subtilis* was 19 mm. Other endophytic fungal extracts may have bioactive compounds in smaller amounts, and it is possible that the crude extracts could give more potent compounds once they have undergone further purification.

## 5. Conclusions

Some isolated endophytes in the study, *P. leucadendri* and *Fusarium* sp. may contain novel bioactive compounds that can be used against tuberculosis. Further studies on cytotoxicity and purification of the compounds still need to be evaluated. The potential of these fungal endophytes is of great interest, but further studies are required.

## Figures and Tables

**Figure 1 ijerph-17-00439-f001:**
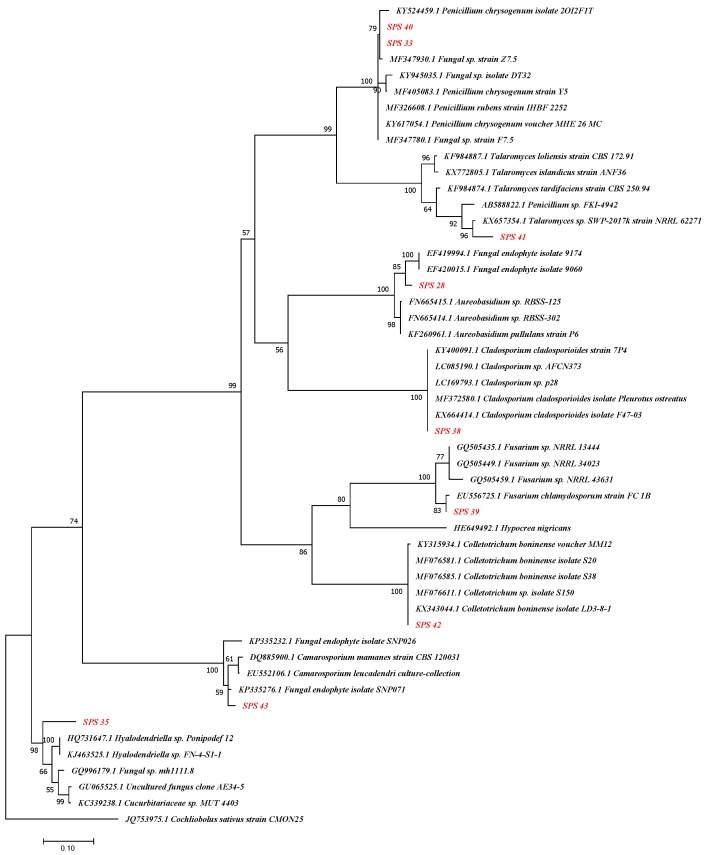
Molecular phylogenetic analysis of ITS1 and ITS4 sequences of all the fungal endophytes isolated from *Solanum mauritianum* alone, with the sequences from NCBI. The analysis was conducted with MEGA 7 using the Maximum Likelihood method.

**Table 1 ijerph-17-00439-t001:** Morphological observations and characteristics of the fungal endophytes.

Isolates	Source	Macro and Microscopic Characteristics	Fungal Endophyte
SPS 28	Stem	Flat cream white colonies which turn black with aging, chlamydospores, colonies are smooth with aging, they become slimy and shiny.	*Aureobasidium pullulans*
SPS 33	Unripe fruit	Powdery green spores on the surface, yellow-white lower surface center.	*Penicillium chrysogenum*
SPS 35	Unripe fruit	Black colonies with a slimy and shiny appearance on the surface.	*Hyalodendriella* sp.
SPS 38	Ripe fruit	Olivaceous green on the top and bottom greenish black, conidiophores which are dense and becoming powdery or velvety due to abundance, reverse of the colony is greenish-black.	*Cladosporium* sp.
SPS 39	Ripe fruit	Colonies grow flat on the media, they are white and turn pink with aging, release orange pigment into the media and spores are formed.	*Fusarium* sp.
SPS 40	Leaves	Darker region at the center, yellow-white lower surface. Powdery green spores on the surface.	*Penicillium chrysogenum*
SPS 41	Leaves	Centre is raised, floccose, sulcate center, bottom orange/white, top orange/yellow, soluble pigment faint orange and as it ages it produces green color spores.	*Talaromyces* sp.
SPS 42	Leaves	White cotton-like colonies turn brown/greenish with age starting at the center, bottom pale yellow or greenish.	*Collectotrichum boninense*
SPS 43	Ripe fruit	They start out as pale green on the plate and later the hyphae become pale brown in color, has hyphae that pale as spores.	*Paracamarosporium leucadendri*

SPS: Sharon Pelo’s Sample, Lactophenol blue dye was used to observe at 40× magnification.

**Table 2 ijerph-17-00439-t002:** List of fungal taxonomic units identified using the molecular identification.

Isolates	Accession No.	Close Relatives	Similarity (%)	Source
SPS 28	MF926050	*Aureobasidium pullulans*	98	Stem
SPS 33	MF928760	*Penicillium chrysogenum*	100	Unripe fruit
SPS 35	MF928761	*Hyalodendriella* sp.	94	Unripe fruit
SPS 38	MF928762	*Cladosporium* sp.	100	Ripe fruit
SPS 39	MF928763	*Fusarium* sp.	99	Ripe fruit
SPS 40	MF928764	*Penicillium chrysogenum*	100	Leaves
SPS 41	MF928765	*Talaromyces* sp.	99	Leaves
SPS 42	MF928766	*Collectotrichum boninense*	100	Leaves
SPS 43	MF928767	*Paracamarosporium leucadendri*	99	Ripe fruit

**Table 3 ijerph-17-00439-t003:** Ethyl acetate crude extract yields from different endophytes.

Fungal Endophytic Isolates	Weight of the Extract (g)
SPS 28	0.34
SPS 33	0.46
SPS 35	0.59
SPS 38	1.02
SPS 39	0.2
SPS 40	1.24
SPS 41	0.54
SPS 43	1.23

Weight of the extracts extracted with ethyl acetate solvent (g) grams.

**Table 4 ijerph-17-00439-t004:** Antimicrobial activity of the fungal endophytes isolated from *Salonum mauritianum*.

Crude Extracts of the Fungal Endophytes			Zone of Inhibition (mm) *				
	Gram Positive		Gram Negative			Acid-Fast Stain	
	*Bs*	*Sa*	*Kp*	*Ec*	*Pa*	*Mb*	*Ms*
*A. pullulans*	7 ± 1.0	9.9 ± 05	10 ± 2.7	13 ± 1.0	9.7 ± 0.6	11 ± 1.0	10.7 ± 0.6
*P. leucadendri*	-	7.3 ± 1.5	9.7 ± 0.6	11.7 ± 0.6	11 ± 1.0	7.7 ± 1.5	13 ± 1.0
*Cladosporium* sp.	10.3 ± 1.5	7.3±0.6	14.7 ± 0.6	8.7 ± 0.6	10.3 ± 0.6	12.7 ± 2.3	9.7 ± 0.6
*Fusarium* sp.	8.7 ± 0.6	9.7 ± 0.6	10.7 ± 1.5	11.7 ± 0.6	23 ± 2.7	14 ± 1.7	-
*Hyalodendriella* sp.	11.3 ± 0.6	7.3 ± 0.6	9.7 ± 0.6	10.7 ± 3.5	9.3 ± 0.6	12.7 ± 2.3	6.7 ± 0.6
*P. chrysogenum* (F)	12.3 ± 1.2	8.7 ± 1.5	-	11.7 ± 0.6	7.7 ± 0.6	14 ± 2.9	10.3 ± 0.6
*P. chrysogenum* (L)	13.7 ± 0.6	8.7 ± 2.3	-	12.3 ± 2.1	6.7 ± 0.6	10 ± 2.9	11.3 ± 2.3
*Talaromyces* sp. (EA)	-	14.7 ± 1.2	-	17.3 ± 2.1	7.7 ± 2.1	8 ± 1.7	9.7 ± 0.6
*Talaromyces* sp. (MeoH)	11 ± 1.0	12.7 ± 2.5	18.3 ± 0.6	19 ± 1.0	9.7 ± 0.6	10.3 ± 0.6	10.3 ± 0.6
Positive control	27 ± 1.7	32.3 ± 0.3	26 ± 1.7	32.6 ± 0.3	26 ± 1.7	27 ± 1.7	27 ± 1.7

*Bs*: *Bacillus subtilis*, *Sa*: *Staphylococcus aureus*, *Kp*: *Klebsiella pneumoniae*, *Ec: Escherichia coli*, *Pa: Pseudomonas aeruginosa*, *Mb: Mycobacterium bovis* and *Ms: Mycobacterium smegmatis*. Positive control: Rifampicin (40 µg/mL). *: mean diameter on zone of inhibition ± Standard Deviation (*n* = 3). -: no zones of inhibition were observed.

**Table 5 ijerph-17-00439-t005:** Minimum inhibitory concentrating assay of the fungal secondary metabolites.

Crude Extracts of the Fungal Endophytes	Minimum Inhibitory Concentration (mg/mL)							
	Gram Positive			Gram Negative			Acid-Fast Stain	
	*Bs*	*Sa*	*Ko*	*Kp*	*Ec*	*Pa*	*Mb*	*Ms*
*A. pullulans*	2.13	2.13	2.13	0.17	2.13	0.17	0.17	0.17
*P. leucadendri*	1.16	2.33	1.16	0.006	1.16	0.006	0.006	2.91
*Cladosporium* sp.	6.37	1.28	6.38	0.049	0.049	0.049	0.049	1.28
*Fusarium* sp.	0.03	5	1.25	0.009	1.25	0.009	0.009	0.009
*Hyalodendriella* sp.	7.38	1.48	1.48	0.03	5	0.03	0.03	0.03
*P. chrysogenum* (*fruit*)	5.75	1.15	5.57	0.02	5	0.02	0.02	0.02
*P. chrysogenum* (*leaves*)	0.75	6	3	0.01	0.01	0.01	0.01	0.01
*Talaromyces* sp.	1.69	3.38	1.35	0.03	0.03	0.03	0.03	0.03
Positive control *	0.0019	0.0019	0.0019	0.0019	0.0019	0.0019	0.0019	0.0019
Negative control ^†^	10	10	10	10	10	10	10	10

*Bs*: *Bacillus subtilis*, *Sa*: *Staphylococcus aureus*, *Ko*: *Klebsiella oxytoca*, *Kp*: *Klebsiella pneumoniae*, *Ec*: *Escherichia coli*, *Pa*: *Pseudomonas aeruginosa*, *Mb*: *Mycobacterium bovis* and *Ms*: *Mycobacterium smegmatis*. Positive control *: Rifampicin (40 µg/mL). Negative control ^†^: DMSO (10%).

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
