# Peer review of "Diversity and Antimicrobial Activity of Culturable Fungal Endophytes in Solanum mauritianum"

_ijerph, 2020, doi:10.3390/ijerph17020439_

Round 1

Reviewer 1 Report

The paper “Diversity and antimicrobial activity of culturable 2 fungal endophytes in Solanum mauritianum”  reports results on the antimicrobial activity of crude extracts from endophytic fungi of the plant species Solanum mauritianum. The paper is worth to be published with minor revision.

Suggestions and corrections are reported as follows:

Abstract

15  “ endophytes were isolated from Solanum mauritianum, identified and classified using morphological” change in “endophytes were isolated from Solanum mauritianum, identified and classified using morphological”

17 ”Eight fungal endophytes were identified Aureobasidium pullulans, Paracamarosporium  leucadendri, Cladosporium sp., Collectotrichum boninense, Fusarium sp., Hyalodendriella sp., Talaromyces sp., and Penicillium chrysogenum (from the leaves and unripe fruits)” change in “ Eight fungal endophytes were isolated from the leaves and unripe fruits  and identified as Aureobasidium pullulans, Paracamarosporium leucadendri, Cladosporium sp., Collectotrichum boninense, Fusarium sp., Hyalodendriella sp., Talaromyces  sp., and Penicillium chrysogenum

Materials and Methods

58 Add the Author to Solanum mauritianum

60 “The plant was chosen based on it being invasive, it grows fast and is” change in “The plant species was chosen based on it being invasive, it grows fast and is”

107   “4.5 Isolation of secondary metabolites” change in “4.5 Preparation of the crude extracts”

133  “2.7 Disc diffusion assay of secondary metabolites from Fungal endophyteschange in2.7 Disc diffusion assay of crude extracts from Fungal endophytes

193 “3.4 Secondary metabolites synthesis” change in “3.4 Yeld of extracts”

Table 3 “weight of the secondary metabolites” change in “weight of the extracts” also at the end of the table

207  and 208  “sp.” change in “sp.” And erase “at” after Fusarium sp.

228 change the figure 2 in a table. The results can be better appreciated. In many cases the text doesn’t report the same  MIC as reported in the figure. All  the MIC reported in the text must be consistent with those in the figure (transform in a table).  Moreover the MIC of DMSO must be reported in the table. Write the exact substance (rifampicin)  for the positive control

Discussion

234-238. Report the entire genus when the species is reported for the first time in the page and abbreviated when is reported again.  i.e.  Mycobacterium bovis and  M. smegmatis;

237 “Solanum mauritianum” change in “S. mauritianum”

240-242 It is better to eliminate this phrase because Colletotrichum and Fusarium are important potential pathogens. They can be in endophytic forms at the begining  and change in pathogenic forms later.

288 “plants origin” change in “plant origin”

298 “May” change in “may”

Reviewer 2 Report

Page 4, Line 156, “40µg/ml” should be “40 µg/ml”. Page 4, Line 158, “30µl” should be “30 µl”. Page 5, Table 2, The similarity of the SPS 35 is 94%, why don’t you choose the higher level? Page 7, Lines 205-207, In Table 4, the largest zone of inhibition should be observed against P. aeruginosa (23±2.7 mm), why do you ruling it out? Page 8, Line 234, “M bovis” should be “M. bovis”. Page 9, Line 298, “May” should be “may”.
